# Reverse Electron Transport at Mitochondrial Complex I in Ischemic Stroke, Aging, and Age-Related Diseases

**DOI:** 10.3390/antiox12040895

**Published:** 2023-04-06

**Authors:** Vishal Chavda, Bingwei Lu

**Affiliations:** Department of Pathology, School of Medicine, Stanford University, Stanford, CA 94305, USA

**Keywords:** mitochondrial complex I, reverse electron transport (RET), reactive oxygen species (ROS), NAD^+^/NADH ratio, RET inhibitor, sirtuin, FOXO, aging, Alzheimer’s disease (AD), cancer

## Abstract

Stroke is one of the leading causes of morbidity and mortality worldwide. A main cause of brain damage by stroke is ischemia-reperfusion (IR) injury due to the increased production of reactive oxygen species (ROS) and energy failure caused by changes in mitochondrial metabolism. Ischemia causes a build-up of succinate in tissues and changes in the mitochondrial NADH: ubiquinone oxidoreductase (complex I) activity that promote reverse electron transfer (RET), in which a portion of the electrons derived from succinate are redirected from ubiquinol along complex I to reach the NADH dehydrogenase module of complex I, where matrix NAD^+^ is converted to NADH and excessive ROS is produced. RET has been shown to play a role in macrophage activation in response to bacterial infection, electron transport chain reorganization in response to changes in the energy supply, and carotid body adaptation to changes in the oxygen levels. In addition to stroke, deregulated RET and RET-generated ROS (RET-ROS) have been implicated in tissue damage during organ transplantation, whereas an RET-induced NAD^+^/NADH ratio decrease has been implicated in aging, age-related neurodegeneration, and cancer. In this review, we provide a historical account of the roles of ROS and oxidative damage in the pathogenesis of ischemic stroke, summarize the latest developments in our understanding of RET biology and RET-associated pathological conditions, and discuss new ways to target ischemic stroke, cancer, aging, and age-related neurodegenerative diseases by modulating RET.

## 1. Introduction

Stroke is the second leading cause of morbidity and mortality worldwideafter cancer. In almost 80% of the cases, stroke develops due to a cerebral artery obstruction and/or occlusion [1,2]. During ischemic stroke, the absence of blood supply deprives the brain cells of the glucose and oxygen nutrients they require, disrupting their cellular homeostasis and ultimately resulting in cell death [2]. Mitochondrial dysfunction and deleterious post-stroke ROS are considered hallmark stroke pathologies [3]. A number of pathophysiological processes, including oxidative stress, excitotoxicity, dysregulated endocrine signaling, inflammation, and apoptosis, are involved in the complex pathophysiological process known as cerebral ischemia/reperfusion injury (CIRI), which frequently causes neuronal injury, cell death, and permanent brain damage [3]. ROS-induced oxidative stress during cerebral ischemia leads to eventual cell death after reperfusion. Mitochondrial ROS (mito-ROS) plays a detrimental role in neuronal death during CIRI at several key stages: inflammation, blood brain barrier (BBB) disruption, mitochondrial respiratory chain complex I-III dysfunction, oedema formation, and apoptosis and autophagy [4,5,6] Currently, thrombolytic and endoscopic thrombectomy are the only therapeutic options, along with post-stroke conservative treatments. The tissue plasminogen activator (tPA) is the only Federal Drug Administration (FDA)-approved therapy for stroke treatment, but due to its various side effects and limited therapeutic window, it only benefits a small portion of stroke patients [7,8,9,10,11]. Hence, there is an urgent need to develop safer and novel therapeutic options for the treatment of ischemia reperfusion injuries. Due to reduced blood flow and hypoxic conditions, mitochondria are heavily affected by the low O_2_ and glucose environment. Since the 1960s, the effect of ischemia reperfusion injury on mitochondria has been a research focus and extensive efforts have been dedicated to developing therapeutic strategies to combat reperfusion injury. In this review, we shed light on the relationship between ischemia reperfusion injuries and mitochondrial dysfunction due to reverse electron transport (RET). We also discuss recent studies on the role of RET in other disease conditions and how these studies may help to understand ischemia-reperfusion injury in stroke and inform future therapeutic developments.

## 2. ROS and Oxidative Stress in Stroke

### 2.1. ROS Generation during Stroke

Oxidative stress is the result of an imbalance between ROS production and antioxidant defense mechanisms in cells [3]. Because of its high and specific metabolic activity, neurons are particularly vulnerable to oxidative damage. High oxygen consumption, almost entirely oxidative phosphorylation, low energy reserves, high concentrations of peroxidizable lipids, and high levels of iron acting as prooxidants all contribute to this vulnerability [12]. As a result, neuronal cells are extremely vulnerable to metabolic/ischemic damage and the associated oxidative stress [13]. The macro- and micromolecular changes of neuronal cells ends in neurodegeneration in various neurological disorders, e.g., post-stroke dementia, Alzheimer’s disease (AD), vascular dementia, and others [14]. The progression of ischemic stroke pathology is closely connected to dysregulated ROS. The most common ROS are superoxide (O_2_^−^), hydrogen peroxide (H_2_O_2_), hydroxyl radical (HO^−^), hypochlorous acid (HOCl^−^), nitric oxide (NO), and peroxynitrite (ONOO^−^) that are produced by either intracellular responses (mitochondria) or extracellular inflammation [3,15]. The intracellular production of ROS is mainly due to the altered metabolic activity of the mitochondrial respiratory chain, whereas extracellularly it is a result of inflammasome activation and the immune response [16,17]. These reactive molecules can cause lipid peroxidation, protein oxidation, and DNA and RNA damage, resulting in cellular homeostatic failure and tissue damage (Figure 1). Nitric oxide synthetase (NOS) produces NO that is instrumental in the immune response; phagocytes produce large amounts of NO during ischemic brain injury [3,18]. There are several natural defense mechanisms in the cell to remove ROS or prevent oxidative damage [19]. These include catalase (CAT), heme oxygenase (HO), glutathione reductase, glutathione (GSH), glutathione peroxidase (GSH-Px), superoxide dismutase (SOD), and Vitamins E and C. Microglia and astrocytes are the primary producers of ROS and reactive nitrogen species (RNS), the latter is produced by endothelial NOS (eNOS), inducible NOS (iNOS), or neuronal NOS (nNOS) during ischemic brain injury, which together influence synaptic transmission as well as non-synaptic communication between neurons and glia [20,21]. ROS and RNS diffuse to the oligodendrocyte myelin sheath during periods of increased neuronal activity, activating the protein kinase C and post-translationally modifying the myelin basic protein, a key structural component of myelin [22,23].

### 2.2. ROS-Induced Damages during Stroke

The consequences of ROS imbalance in ischemic stroke are significant, and include apoptosis, disruption of the BBB, inflammation, edema formation, autophagy, and other pathophysiological events. O_2_, H_2_O_2_, and NO play critical roles in neuron-glia communication in the hippocampus [24]. The synaptic long-term potentiation (LTP), which is necessary for memory formation in the hippocampus, also becomes impaired by excessive ROS production during ischemia, resulting in post-stroke cognitive decline [25,26]. ROS-mediated injury can result in the formation of conjugated dienic hydroperoxides, which can be degraded into aldehydes, dienals, or alkanes that are extremely toxic to neurons and white matter, resulting in apoptosis followed by chronic neurodegeneration after ischemic brain damage [27]. The inflammation and oxidative stress that develop in the brain after a stroke have been linked more recently to tryptophan oxidation via the kynurenine pathway [28]. The release of glutamate is a crucial mechanism that determines tissue damage after cerebral ischemia [29]. On the other hand, when the a-amino-3-hydroxy-5-methyl-4-isoxazolepropionic acid (AMPA) receptors are activated, oxygen is produced. When oxygen combines with NO, it creates the extremely harmful ONOO. When one of the ROS species is inactivated, the vessel can expand due to the other reactant (ROS) escaping [30]. It has also been shown that increased kynurenic acid levels are linked to poorer outcomes, and the infarct volume is strongly correlated with a decreased ratio of 3-hydroxyanthranilic acid to anthranilic acid (a free radical generator) [31,32]. Nuclear DNA damage has been linked to two distinct mechanisms, including DNA fragmentation caused by endonucleases and oxidative modification. In ischemic stroke injury, poly (ADP-ribose) polymerase (PARP) activation occurs in two phases, starting in the neuronal components and localizing 3–4 days later in the infiltrating inflammatory cells [33]. It has also been suggested that there may be reductions in the nuclear protein apurinic apyrimidinic endonuclease (APE/Ref-1), which cleaves ROS-induced apyrimidinic sites in oxidized DNA [27]. Moreover, following transient global ischemia, the p53-upregulated modulator of apoptosis (PUMA) is upregulated in the hippocampal neurons [34]. ROS can also activate caspase-activated DNase (CAD), which cleaves DNA and causes apoptosis, resulting in progressive neurodegeneration and associated post-stroke disorders [35]. Thus, excessive ROS production can cause mitochondrial dysfunction leading to overall cellular dysfunction and cell death in ischemic stroke (Figure 1).

## 3. Mitochondrial Dysfunction and Ischemic Stroke

The first sign of an ischemic stroke is cerebral artery occlusion, which is followed by severe glucose and oxygen deprivation (OGD) and a number of pathological processes that ultimately result in irreversible brain damage and neuronal death [36]. Ischemic stroke may cause excessive accumulation of Ca^2+^ ions inside the mitochondria, which can lead to an imbalance of fission and fusion dynamics, mitochondrial dysfunction, mitochondrial-induced apoptosis, and mitophagy [37,38]. Indeed, a number of studies have demonstrated that mitochondrial dysfunction contributes to the pathology of ischemic stroke, which arises from the abnormal ROS production, calcium buildup, defective mitochondrial biogenesis, activation of apoptosis, disruption of ATP: ADP ratios, and reduction in NAD^+^ levels [39,40,41]. The mitochondrial dysfunction can trigger apoptosis by increasing the generation of ROS, calcium accumulation, and cytochrome c release [42]. Moreover, inflammatory responses may be activated during ischemic stroke, mediated by the NOD-like receptor protein 3 (NLRP3)-inflammasome [43,44]. As a result of OGD and cerebral IR injury, mitochondrial dysfunction has been demonstrated to be pivotal in the activation of the NLRP3 inflammasome in microglia. According to one study, mitochondrial dysfunction caused by the opening of the mitochondrial permeability transition pore (MPTP) resulted in an increase in ROS generation, which facilitated the activation of the NLRP3 inflammasome that is linked to abnormal mitochondrial fusion and fission events [45,46]. Mitochondrial dysfunction during stroke is also associated with multiple proteomic changes present in complex I proteins (e.g., NDUFS3, V1, V2, S1, S2). Such changes also occur during aging and age-related neurodegeneration [47], and may be responsible for RET and RET-ROS production during ischemic reperfusion (see later). Mitochondrial complex I, the first protein complex of the mitochondrial respiratory chain (MRC) that is also known as NADH dehydrogenase, is potentially involved in various neurodegenerative disorders through excessive ROS generation and modulating membrane polarization [48,49]. Though succinate dehydrogenase (complex II) is less involved in MRC defects during ischemia, it, nevertheless, also plays a role in the induction of ROS production and apoptotic cell death [17,50]. Such apoptotic pathways are characterized by intrinsic and extrinsic pathways of mitochondrial events during ischemic injury, e.g., loss of mitochondrial membrane potential, modifications of MRC, cytochrome *c* release, and impaired cellular redox state or abnormal antioxidative defense mechanism [51].

## 4. Mitochondrial Complex I Dysfunction in Ischemia Reperfusion Injury

In the middle of the 1960s, the work of Ozawa and colleagues [52,53,54] demonstrated the dramatic effects of oxygen deprivation on mitochondrial metabolism in the brain and established the effect of brain ischemia on mitochondrial respiration. Ozawa and colleagues also reported that during ischemia with no reperfusion conditions, isolated mitochondrial fractions exhibited hampered glutamate-dependent respiration. Ginsberg and colleagues reported that the functions of mitochondrial complex I, ubiquinol-cytochrome c oxidoreductase (complex III, or cytochrome bc1 complex), and cytochrome c oxidase (complex IV) declined dramatically in the gerbil (Mongolian rodent) brain after ischemia [55]. Moreover, in 1978, Nordstrom and colleagues reported that complete and incomplete ischemia have different effects on mitochondrial complex I-supported activity. They reported that incomplete ischemia can cause more dramatic decline in malate- and glutamate-driven respiration just after 1 h of ischemia reperfusion [56]. Allen and colleagues performed the first direct experiment showing the effects of ischemia reperfusion injury on the mitochondria and complex-I respiration on gerbil by carotid artery occlusion [57]. Unlike the widely used rodents, gerbils do not have the circle of Willis and, thus, offer an advantage for modelling human global and focal cerebral ischemia. Yoshimoto and colleagues reported that the complex I activity was initially decreased after 2 h of reperfusion but then gradually increased at 4 h post-reperfusion [58]. Niatsetskaya et al. confirmed similar results of ischemia on mitochondrial dysfunction but later studies also reported that malate-glutamate supported mitochondrial respiration which improved just after reperfusion significantly declined after 4 h of ischemia-reperfusion [59]. In a monkey model of 3 h of Middle Cerebral Artery Occlusion (MCAO), Tsukada and colleagues reported a consistent decline in mitochondrial complex 1 activity through positron emission tomography assay [60]. Thus, ischemia reperfusion injury and mitochondrial complex I activity collapse are frequently reported coexisting post-ischemic cellular events, but the mechanism of mitochondrial complex I inhibition is still poorly understood [61,62].

The proton motive force (PMF; also known as the electrochemical gradient, Δp) across the inner mitochondrial membrane is produced by the proton pumping through complex I, complex III, and complex IV. The PMF is composed of a membrane potential and a proton gradient (ΔpH) (ΔW). Chemiosmosis allows the protons to return to the matrix where ATP is produced. The forward electron flux in the electron transport chain (ETC), which is mediated by an iron-sulfur (Fe-S) cluster containing proteins, has an inverse relationship with the electron slip, ΔpH, and ΔW. In general, for FET to happen, the difference in the redox potential between the NAD^+^/NADH and CoQ pool across complex I (ΔE_h_) needs to be enough to pump protons against the ΔpH (2 ΔE_h_ > 4 ΔpH, as 4 protons are pumped for every 2 electrons that pass complex I). Under conditions of 2 ΔE_h_ < 4 ΔpH, RET will happen. Pathophysiological conditions (ischemic stroke or reperfusion) with low PO2 and low ADP-Pi/ATP ratios (high NADH/NAD^+^ and FADH2/FAD ratios) can slow forward electron flux and increase membrane potential for a given reductive pressure. Less consumption of PMF by ATP synthase (complex V) and more electron transport toward complex I will occur. This causes an imbalance between RET and FET in favor of RET and producesmany ROS [63,64,65,66]. Previous studies concluded that mitochondria lose their inner membrane and become fragmented followed by swelling in just a couple of minutes due to low oxygen (hypoxia) after ischemia reperfusion injury [67]. Studies also reported, based on triphenyltetrazoliumchloride (TTC) staining of various ischemic brain samples, that NADH-dependent dehydrogenases became impaired after ischemia prior to succinate dependent activity change [67,68].

## 5. RET in Ischemic Stroke

Ischemic stroke is the result of blocked blood flow and oxygen to particular parts of the brain, resulting in bioenergetic failure of the mitochondria [69]. During FET in mitochondrial complex I, ubiquinone (CoQ10) oxidizes the NADH produced by the tricarboxylic acid (TCA) cycle, generating ubiquinol (CoQ10H2) and NAD^+^. This conversion induces free electrons that are accepted by flavin mononucleotide (FMN) and further transferred sequentially to CoQ10, cytochrome C, and, ultimately, to O_2_ to generate H_2_O by complex IV. In the presence of sufficient mitochondrial membrane potential, the enzyme can operate in reverse to conduct RET by catalyzing the reduction of matrix NAD^+^ with electrons provided from ubiquinol. During ischemic stroke, there is a dramatic accumulation of succinate relative to other metabolites [70] The electrons from succinate are transferred to ubiquinone, some of which are further moved to complex-III (upstream, FET) and others flow back to complex I (downstream, RET). Excessive ROS is generated during RET compared to FET, presumably due to the dissociation of FMNH2 from complex I and the reaction with matrix O_2_ [62]. Studies support both the FMN and the CoQ reduction sites of mitochondrial complex I as possible centers for ROS production during RET [71,72]. During ischemia and reperfusion, due to activated RET, there is a burst of up-regulated ROS [73]. This RET-ROS is proposed to be responsible for potential mitochondrial dysfunction during IR injury [61,62] Reperfusion is an event when the proton-motive force is rapidly increased due to the re-oxygenation of mitochondria within a few seconds of blood reflow and is needed for cellular survival after ischemia [74]. This hypothesis of IR injury is supported by previous experiments confirming the proton-motive force (∆p) as a driver of RET and RET as a major mechanism for mito-ROS generation [75,76,77] The burst of RET-ROS generation may also be a major cause of complex I dysfunction during post-ischemia reperfusion injury [77]. During ischemia and post-ischemia reperfusion, the NAD^+^/NADH ratio may also be dramatically reduced due to highly up-regulated RET. Whether excessive RET-ROS production and reduction in the NAD^+^/NADH ratio both contribute to IR injury in stroke remains to be tested. The inhibition of RET with small molecules has shown efficacy in preventing ROS overproduction, preserving mitochondrial function, and ameliorating stroke damage in animal models [61,62]. This offers hope that RET inhibitors are potential therapeutics for preventing ischemia-reperfusion injuries.

## 6. RET in Cancer, Aging, and Age-Related Neurodegenerative Diseases Such as AD

The brain consumes higher amounts of energy and exhibits a higher metabolic rate [78]. In a resting awake state, the brain accounts for 2% of the total body weight but consumes 25% of the total body glucose and 20% of the total body oxygen consumption [79]. As mitochondria play a critical role in oxidative and energy metabolism, calcium homeostasis, and cell survival [80,81], mitochondrial dysfunction is intimately linked to brain diseases. Gene expression studies in AD repeatedly identified defects in mitochondria-related metabolic pathways, providing direct evidence for mitochondrial impairment in AD. In both early and definite AD brain specimens, complex I was down-regulated, whereas complexes III and IV showed increased mRNA expression [82,83]. Recent research found that mild inhibition of complex I by a cell-permeable tricyclic pyrone CP2 increased the respiratory capacity and reduced proton leaks [84]. CP2 has been shown in animal models of AD to alleviate cognitive and pathological deficits, and it is also being developed to treat AD, indicating the potential use of complex I modulators in AD treatment [84,85]. In a clinical report, ETC complexIII and IV were found to be impaired in human AD brain samples of the cerebral cortex, temporal cortex, and hippocampus [75]. Compromised complexIII and IV functions could lead to increased RET and contribute to AD pathology, although this remains to be directly demonstrated using clinical samples. However, recent studies in *Drosophila* models of AD expressing full-length amyloid precursor proteins (APPs) or the C-terminal fragment of APP (APP.C99), or in human induced pluripotent stem cell (iPSC)-derived neuronal models of AD, showed that RET was activated, as demonstrated by the increased production of RET-ROS and decreased NAD^+^/NADH ratio [47].

Aging and age-related diseases are assumed to share some basic biological mechanisms, but this has not been demonstrated comprehensively. Along with the loss of proteostasis, cellular senescence, deregulated nutrient sensing, stem cell exhaustion, epigenetic alteration, genome instability, telomere attrition, and altered intercellular communication, mitochondrial dysfunction is regarded as one of the fundamental hallmarks of aging [86]. Given that increased ROS and decreased NAD^+^ levels and, thus, a lowered NAD^+^/NADH ratio, the key outcomes of RET, are hallmarks of aging, it is possible that RET is activated during normal aging. Indeed, this was found to be the case in *Drosophila* [47]. Moreover, in a *Drosophila* model of a Notch-induced brain tumor [87,88], RET was also found to be elevated. Intriguingly, mechanistic studies in the *Drosophila* brain tumor model demonstrated that the Notch protein acts in a non-canonical manner by entering the mitochondria and directly interacting with complex I subunits involved in RET, including the NDUFS3 and NDUFV1, altering the dynamics of the protein-protein interactions involved in RET [89]. These data suggest that mitochondrial complex I involved in RET may adopt a different conformation than the complex I involved in FET. Similar protein-protein interaction changes were observed during normal *Drosophila* aging [47], suggesting a general mechanism of RET deregulation. Notch, therefore, becomes the first protein not belonging to the ETC to be recruited to complex I and directly participate in RET. It is worth noting that, in addition to its role in neural development, neural stem cell biology, and cancer stem cell biology, Notch has been linked to aging, AD, and stroke [90,91]. It will be an interesting future direction to test if Notch and its interaction with complex I proteins are involved in RET deregulation during ischemic stroke. It will also be interesting to identify other non-ETC proteins that are recruited to complex I to influence RET under other pathophysiological conditions.

## 7. RET as a Therapeutic Target for Stroke, Cancer, Aging, and Age-Related Disorders

Increased ROS and decreased NAD^+^ levels and thus, a lowered NAD^+^/NADH ratio are key parameters associated with aging and age-related diseases. The implication of RET in these conditions raises the interesting possibility that RET is a major contributor, and that the inhibition of RET may be beneficial in the settings of stroke, cancer, aging, and age-related diseases. 

### 7.1. RET Inhibition in Stroke

In stroke, complex I inhibitors have been shown to protect against I/R injury [62,92]. The protective effects of many complex I inhibitors previously shown in I/R settings are possibly mediated by their inhibitory effect on RET. The manipulation of succinate accumulation or oxidation through SDH inhibition has also been shown to protect against I/R injury [70]. However, the non-selectivity of these inhibitors on RET and FET carries significant risk for side effects in the clinic. A promising alternative approach to preventing I/R injury is to take advantage of the active/deactive transition of complex I during I/R [93] by targeting Cys residues exposed in the deactive state through reversible thiol modification [94] This temporarily locks complex I in a state non-conducive to RET, thus reducing RET-induced damages [70,94]. The advantage of this approach is that it does not affect active complex I and that the thiol lock can be removed by endogenous antioxidants. However, such thiol-reacting compounds must be applied at the time of reperfusion, as delivery even 10 min later has proven to be ineffective [94]. Therefore, it remains uncertain whether the current agents are capable of rescuing the secondary mitochondrial failure that occurs hours later and is known to be critical for I/R injury [62].

### 7.2. RET Inhibition in Aging and Age-Related Diseases

In aphenotype-based screen, 6-chloro-3-(2,4-dichloro-5-methoxyphenyl)-2-mecapto-7-methoxyquinazolin-4(3H) (CPT) was identified as a novel inhibitor of a Notch-induced brain tumor that acts on RET [89]. CPT specifically binds to NDUFS3 of complex I, altering the NDUFS3 interaction with Notch and other proteins in the matrix arm of complex I, and inhibits RET-induced ROS production and NAD^+^/NADH ratio decrease [89]. In animal models, RET is shown to be activated during normal aging, in the context of Notch-induced brain tumor [89], and in AD. RET inhibition by CPT treatment extended the lifespan and rescued the brain tumor and AD phenotypes in these settings. Moreover, the partial knockdown of NDUFS3 has similar protective effects as CPT in the contexts of aging, brain tumor, and AD, and CPT treatment in the NDUFS3 knockdown condition did not offer additional benefits [47,89], supporting the assertion thatCPT specifically engages with RET through NDUFS3 in vivo to offer its biological benefits. Importantly, the effect of CPT in aging, age-related AD pathogenesis, and brain tumor settings can be faithfully recapitulated by the supplementation of the NAD^+^ precursors of nicotinamide mononucleotide (NMN) and nicotinamide riboside (NR), supporting the notion that the NAD^+^/NADH ratio is a key mediator of the biological effects of RET inhibition by CPT. At least in cancer settings, RET-ROS was found to not be critical for the effects of RET and CPT on cancer stem cell proliferation [89]. It remains to be determined whether RET-ROS is important for the biological effects of RET in aging and age-related diseases. Interestingly, whereas a previous study proposed that RET-ROS helps promote the lifespan in *Drosophila* [95], the supplementation of a mitochondria-targeted antioxidant mito-TEMPO extended the fly lifespan [47], suggesting that mito-ROS in general is detrimental to the lifespan, consistent with many previous studies in diverse organisms [96]. Thus, the role of various sources of mito-ROS in lifespan regulation is complex and requires further investigation. 

### 7.3. RET Inhibition on the NAD^+^-Dependent Sirtuin/FOXO/Autophagy Pathway

Mitochondria are critical for NADH/NAD^+^ metabolism. During the TCA cycle, NAD^+^ gains two electrons and a proton to be reduced to NADH. NADH is in turn oxidized to NAD^+^ upon donating electrons to NADH dehydrogenase (complex I) of the ETC. The central roles of NAD^+^ and NADH in TCA and ETC, respectively, underscore the importance of balanced NAD^+^/NADH to mitochondrial function [97]. Beyond this central metabolic function, NAD^+^ critically regulates the activities of NAD^+^-consuming enzymes, including sirtuins, poly-ADP-ribose polymerases, and CD38/157 ectoenzymes, which have been implicated in the aging process [98,99,100,101] NAD^+^/sirtuin modulates longevity through the activation of mitochondrial stress responses such as the unfolded protein response (UPR^mt^) and the nuclear translocation and activation of the FOXO transcription factor [102], providing a molecular link between mitochondrial stress signaling and longevity. It has been shown that the NAD^+^ level declines with age [102,103,104], and that replenishing NAD^+^ with a NAD^+^ precursor offers beneficial effects towards aging and age-related disorders [105]. However, the causes of NAD^+^ depletion during aging are not well-defined. The activation of RET is emerging as a key mechanism of NAD^+^ depletion and the attenuation of NAD^+^-regulated lifespan regulatory activities during aging. Indeed, the lifespan extension by CPT-mediated RET inhibition is dependent on the sirtuin/FOXO/autophagy pathway [47] (Figure 2). It will be exciting to test if the sirtuin/FOXO/autophagy pathway is also involved in the beneficial effects of RET inhibition in ischemic stroke and other disease settings where mitochondrial complex I has been implicated.

## 8. Conclusions

Ischemic stroke is a disease with limited therapeutic options and a high rate of morbidity and mortality. Previous studies on stroke and ischemia-reperfusion injury have mainly focused on ROS, but the main source of ROS generation is poorly defined, and antioxidant therapeutics have largely failed in the clinic. With the new understanding of RET being a main source of pathological ROS during ischemic stroke, cancer, and age-related diseases, and the RET-induced NAD^+^/NADH ratio decrease being another important mediator of the biological effects of RET, new therapeutic approaches targeting the RET process or the downstream effectors of RET—both RET-ROS and the NAD^+^/NADH imbalance—are warranted for the treatment of ischemic stroke as well as other disease conditions where RET has been implicated, from cancer to age-related neurodegenerative diseases.

## Figures and Tables

**Figure 1 antioxidants-12-00895-f001:**
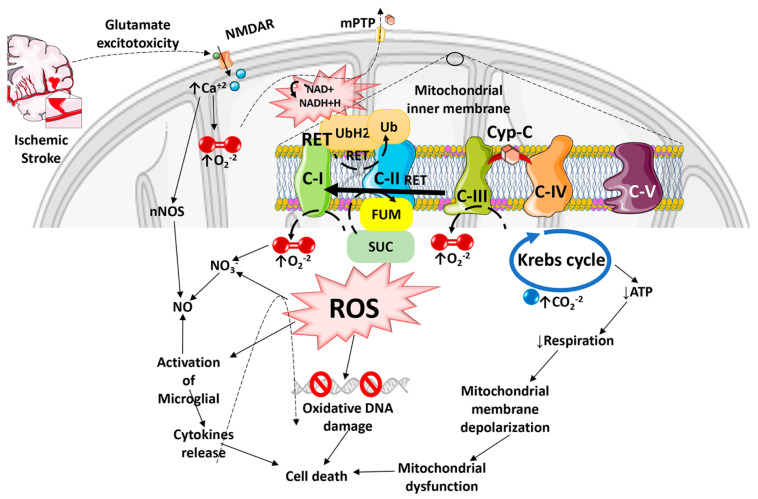
Diagram illustrating how Reactive Oxygen Species especially Reactive Oxygen Species generated by the Reverse Electron Transport process, and the ensuing oxidative stress contributes to mitochondrial dysfunction, cell death, and progressive ischemic stroke pathology. ROS can damage mitochondrial DNA because of the lack of a chromatin-like structure that would protect DNA against ROS insults. ROS can also damage lipids and protein structures in mitochondrial matrix and further exacerbate mitochondrial dysfunction. Mitochondrial ROS released into the cytosol can activate microglia and astrocytes, causing neuroinflammation and death of injured neurons.

**Figure 2 antioxidants-12-00895-f002:**
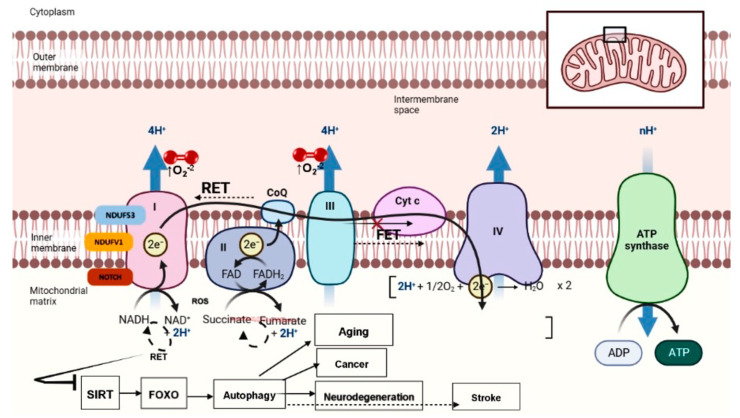
Diagram illustrating RET and RET-regulated NAD^+^-dependent sirtuin signaling pathway as a contributor and potential therapeutic target for aging, age-related diseases, and ischemic stroke. Lack of O_2_ and accumulation of succinate as in ischemic stroke, or impairment of C-III, IV, or V as in aging and age-related diseases can lead to a slowdown of FET and acceleration of RET. In addition to damage by RET-generated ROS (RET-ROS), RET-induced drop of NAD^+^/NADH ratio can compromise the NAD^+^-dependent sirtuin-FOXO-autophagy signaling pathway and contribute to general aging, age-related diseases, including cancer and neurodegeneration, and stroke.

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
