# Peer review of "Reverse Electron Transport at Mitochondrial Complex I in Ischemic Stroke, Aging, and Age-Related Diseases"

_antioxidants, 2023, doi:10.3390/antiox12040895_

Round 1

Reviewer 1 Report

Very dense and comprehensive minireview. It is difficult to follow.

Using subsections would help the understanding.

line 56 may help to understand

line 178, please give some sentences about the gerbils' cerebral circulation. It is not similar to human....

line 250 mitochondria-related

Please define what are complex-I, II, III and IV in the inner mitochondrial membrane.

Please add a chemical equation about RET

The Figure captions are too condensed. Please give a bit more detailed explanation to the illustrations.

Author Response

Very dense and comprehensive minireview. It is difficult to follow.

Using subsections would help the understanding.

Reply: Thanks for the suggestion. We have used subsections.

line 56 may help to understand

Reply: We have corrected the typo

line 178, please give some sentences about the gerbils' cerebral circulation. It is not similar to human....

Reply: We have explained as follows: Unlike the widely used rodents, gerbils do not have the circle 
of Willis and thus offer an advantage for modelling global and focal cerebral ischemia

line 250 mitochondria-related

Reply:We have corrected the typo.

Please define what are complex-I, II, III and IV in the inner mitochondrial membrane.

Reply: We have now defined complex-I, II, III, IV, and V when they first appear as follows: NADH: 
ubiquinone oxidoreductase (complex I), succinate dehydrogenase (complex II), ubiquinolcytochrome c oxidoreductase (complex III, or cytochrome bc1 complex), cytochrome c oxidase  (complex IV), and ATP synthase (complex V).

Please add a chemical equation about RET

Reply: We have added the following chemical equation about FET and RET on page 5, line 199-202

The Figure captions are too condensed. Please give a bit more detailed explanation to the illustrations.

Reply: We have now added more details to the figure captions

Reviewer 2 Report

Overview of the manuscript
The manuscript is a review focused on the
historical account of the roles of ROS and oxidative damage in the pathogenesis of ischemic stroke. The latest developments in understanding the RET biology and RET- associated pathological conditions are presented and discussed.

 GENERAL COMMENT

The importance of ROS-related pathways in determining a worsening of pathological condition in ischemia- reperfusion injury is at today an current issue, to which recent research on oxidant pathways has offered new biological aspect to considerer.  The review is well focused on the subject matter and the bibliography is rich, appropriate and enough current to support to the topic presented. Some points should be better explained.

SPECIFIC COMMENTS

Pag. 3, line 108-111: the sentence should be explained more clearly. Rephrased it.

Pag. 4, line 147: explicit the acronym.

Pag. 4-5, line 177-178: the indication about the gerbil is inappropriate. Gerbil has an incomplete arterial Willis circle. Explain better the issue.

Pag. 6, line 254: explicit the acronym.

Author Response

GENERAL COMMENT

The importance of ROS-related pathways in determining a worsening of pathological condition in ischemia- reperfusion injury is at today an current issue, to which recent research on oxidant pathways has offered new biological aspect to considerer.  The review is well focused on the subject matter and the bibliography is rich, appropriate and enough current to support to the topic presented. Some points should be better explained.

Reply: We thank the reviewer for the positive comments on our manuscript. Some points are now  better explained in the manuscript

SPECIFIC COMMENTS

Pag. 3, line 108-111: the sentence should be explained more clearly. Rephrased it.

Reply: We have rephrased the sentence as follows: It has also been shown that increased kynurenic  acid levels are linked to poorer outcomes, and infarct volume is strongly correlated with a decreased ratio of 3-hydroxyanthranilic acid to anthranilic acid (a free radical generator) (Stone & Darlington, 2002)

Pag. 4, line 147: explicit the acronym.

Reply: We have explained the acronym as follows: Mitochondrial permeability transition pore (MPTP)

Pag. 4-5, line 177-178: the indication about the gerbil is inappropriate. Gerbil has an incomplete arterial Willis circle. Explain better the issue.

Reply: Thanks for pointing this out. We have explained as follows: Unlike the widely used rodents,  gerbils do not have the circle of Willis and thus offer an advantage for modelling human global and focal cerebral ischemia

Pag. 6, line 254: explicit the acronym.

Reply: We have defined CP2, the name of a tested chemical, as cell-permeable tricyclic pyrone CP2